# Novel Bi-Functional 14-mer Peptides with Both Ovarian Carcinoma Cells Targeting and Magnetic Fe_3_O_4_ Nanoparticles Affinity

**DOI:** 10.3390/ma12050755

**Published:** 2019-03-05

**Authors:** Yueting Li, Guangfu Yin, Ximing Pu, Xianchun Chen, Xiaoming Liao, Zhongbing Huang

**Affiliations:** College of Materials Science and Engineering, Sichuan University, Chengdu 610064, China; zoe-lyt@foxmail.com (Y.L.); puximing@163.com (X.P.); kfwcxc@163.com (X.C.); Liao_xm@scu.edu.cn (X.L.); zbhuang@scu.edu.cn (Z.H.)

**Keywords:** bi-functional peptide, magnetic Fe_3_O_4_ nanoparticles, surface modification, tumor targeting, ovarian carcinoma

## Abstract

Fe_3_O_4_ magnetic nanoparticles (Fe_3_O_4_-MNPs) have attracted much interest for their potential medical applications due to their desirable magnetic properties. However, their potential cytotoxicity, high RES clearance in circulation, and nonspecific distribution in tissue might be the main obstacles in practice. In the present study, a novel bi-functional 14-mer peptide with both ovarian carcinoma cells targeting and magnetic Fe_3_O_4_ nanoparticles affinity was designed and synthesized, and then a facile and effective modification method was developed to bestow the Fe_3_O_4_-MNPs with tumor-targeting capability via modification, using the bi-functional peptides. First, on the basis of a tumor-targeting 7-mer peptide QQTNWSL (Q-L) and another Fe_3_O_4_-MNPs-targeting 7-mer peptide TVNFKLY (T-Y)—screened by phage-displayed peptide libraries—two bi-functional 14-mer peptides sequenced as LSWNTQQ-YLKFNVT (abbreviated as LQ-YT) and QQTNWSL-YLKFNVT (QL-YT) were synthesized through combining the Q-L peptide and T-Y peptide in predetermined configurations. Their specificity for bonding with A2780 tumor cells and affinity for Fe_3_O_4_-MNPs were verified. Then the bi-functional 14-mer peptides were applied to modify the Fe_3_O_4_-MNPs. Results showed that both bi-functional 14-mer peptides could be conjugated to the Fe_3_O_4_-MNPs surface with high affinity. Immunofluorescence and Prussian blue staining assays indicated that the LQ-YT-modified Fe_3_O_4_-MNPs could specifically bond to A2780 tumor cells. In addition to our findings suggesting that more β-turns and random coils are conducive to increasing polypeptide surface area for binding and exposing the target group and bonding sites on LQ-YT to external targets, we demonstrated that the bi-functional 14-mer peptide has affinity for Fe_3_O_4_-MNPs, and that Fe_3_O_4_-MNPs, which was modified with a 14-mer peptide, could be bestowed with a targeting affinity for ovarian carcinoma cells.

## 1. Introduction

Internationally, ovarian cancer is the seventh leading cancer diagnosis and eighth leading cause of cancer mortality among women [1]. It is the most lethal gynecologic malignancy [2] and has become still more known for its marked tissue heterogeneity [3]. While chemotherapy and cytoreductive surgery have exhibited favorable curative outcomes for early-stage ovarian cancer, unfortunately more than 75% of cases advance to stage III or IV. Furthermore, the overall five-year survival rate for advanced stage patients is just 20–30% [4]. The lack of effective means for early diagnoses and the deficiency of specific targeting in chemotherapy have been regarded as the main barriers to improving treatment response and cure rates for ovarian cancer patients [5,6].

Owing to their attractive magnetic properties, Fe_3_O_4_ magnetic nanoparticles (Fe_3_O_4_-MNPs) have attracted interest for their potential biomedical applications, including for tumor diagnosis and therapy, for example, as a key contrast agent component for magnetic resonance imaging, as magneto-thermal media for magnetic hyperthermia therapy, or as a drug delivery guiding medium [7,8,9,10,11]. As functional media, besides the desirable magnetic properties, it is essential that Fe_3_O_4_-MNPs exhibit low cytotoxicity, long blood circulation time, and—most importantly—specific affinity for tumors [11,12,13,14].

For clinical tumor-targeting applications, Fe_3_O_4_-MNPs could be concentrated in the focal zone via passive or active targeting strategies. Passive targeted delivery was achieved via macrophage phagocytosis associated with affinity diversity in tissues with different physiological features, resulting in histologically nonspecific distribution, which usually facilitates delivery to the liver, spleen, and lungs [15,16]. Active targeted delivery was actualized by bonding the particles to specifically conjugated carbohydrate–lectin, antibody–antigen, or ligand–receptor moieties in which the lectin, antigen, or receptor either exists uniquely or is over-expressed on the surfaces of tumor cells. The modified particles would differentially recognize and specifically bond to corresponding sites and therefore could be selectively concentrated in neoplastic tissue [17]. Representative targeting moieties include carbohydrates, antibodies, peptides, and some small molecules. Among these potential targeting moieties, peptides stand out as the most attractive options because of their smaller size, higher stability, bonding potential to the tumor cells, and lower immunogenicity. More importantly, the selected functional peptides could specifically target the designated tumor cells or vessel endothelial cells [18,19]. To date, specific targeting peptides for more than 10 types of cancer—including cancer of the stomach, liver, breast, and prostate, among others—have been successfully identified, often in the case of unknown receptor characteristics. However, the modifications for targeting peptides usually require polymers wrapped around the Fe_3_O_4_-MNPs as an intermediate step, which results in a complicated modification and easily aggregated (and thus dysfunctional) modified particles [19,20,21,22,23,24].

In our previous studies, we selected two 7-mer peptides sequenced as QQTNWSL (abbreviated as Q-L peptide) and TVNFKLY (T-Y peptide) via the biopanning of phage-displayed peptide libraries (Ph.D.™-7), with A2780 (human epithelial ovarian carcinoma cell lines) and Fe_3_O_4_-MNPs as the panning objects, respectively. We identified the targeting affinity of A2780 tumor cells for the Q-L peptide and the high affinity of Fe_3_O_4_-MNPs for the T-Y peptide. Additionally, both 7-mer peptides were verified to be cytocompatible with HUVECs and L929 cells [25].

Based on the results of previous studies, this current work was devoted to developing a facile and effective method to modify Fe_3_O_4_-MNPs using a bi-functional peptide with both tumor-targeting ability and Fe_3_O_4_-MNPs affinity. First, several 14-mer peptides were designed and synthesized by combining a 7-mer Q-L peptide and a T-Y peptide in predetermined settings. The 14-mer peptides were intended to exhibit high affinity for Fe_3_O_4_-MNPs and high targeting affinity for A2780 tumor cells. Then, the compositions and secondary structures of the synthesized 14-mer peptides were analyzed. Moreover, bi-functional peptides were used to modify Fe_3_O_4_-MNPs via facile co-incubation, and the features and performances of the modified Fe_3_O_4_-MNPs were systemically characterized, particularly regarding their bonding ability and the extent of their targeting affinity for A2780 tumor cells. Furthermore, the effects of modified Fe_3_O_4_-MNPs on the cellular behavior of HUVECs, L929 cells, and A2780 tumor cells were assessed. The results supported the observation that the bi-functional 14-mer polypeptide LSWNTQQ-YLKFNVT has targeting affinity for ovarian carcinoma cells and the affinity to magnetic Fe_3_O_4_-MNPs.

## 2. Materials and Methods

### 2.1. Materials and Cell Culture

The Fe_3_O_4_-MNPs were purchased from Nanjing Emperor Nano Material Co. Ltd. (Nanjing, China). The designed bi-functional 14-mer peptides with a purity >95%, including the LQ-YT peptide and QL-YT peptide, as well as the corresponding fluorescein isothiocyanate (FITC)-labeled peptides, FITC-LQ-YT and FITC-QL-YT, were synthesized by Shanghai Bootech Bioscience and Technology Co. Ltd. (Shanghai, China) according to our specifications.

A2780 (human epithelial ovarian carcinoma cell line), L929 (mouse fibroblasts cell line), and HUVECs (human umbilical vein endothelial cells) were supplied by the Key Laboratory of Transplant Engineering and Immunology in Sichuan University (Chengdu, China). The cells were washed twice with sterile PBS, and then suspended in Roswell Park Memorial Institute (RPMI) 1640 medium (GIBCO Invitrogen, Carlsbad, CA, USA), supplemented with 10% new calf serum (NBS, Hyclone, Logan, Utah, USA). Cells, propagated as a monolayer culture, were trypsinized every other day. All cultures were incubated at 37 °C under 5% CO_2_ in a saturated-humidity atmosphere.

### 2.2. Design, Prediction, and Characterization of Bi-Functional Peptides

The Fe_3_O_4_-MNPs affinitive 7-mer peptide TVNFKLY was selected to combine with the A2780 tumor cells-targeting 7-mer peptide QQTNWSL (both previously screened by our research group) to construct bi-functional 14-mer peptides, intended to exhibit high affinity for Fe_3_O_4_-MNPs and targeting affinity for A2780 tumor cells. According to different combinations of C-terminals and N-terminals, there were four potential 14-mer peptide configurations: LSWNTQQ-YLKFNVT (abbreviated as LQ-YT) as well as QQTNMSL-YLKFNVT (QL-YT), LSWNTQQ-TVNFKLY (LQ-TY), and QQTNMSL-TVNFKLY (QL-TY). In our previous work, LQ-TY peptide was found to have a much higher affinity for Fe_3_O_4_-MNPs than QL-TY, but it conjugated with A2780 tumor cells with a high affinity but not specificity [26]. Both LQ-YT and QL-YT were synthesized and investigated in this study.

The Protean tool of DNAStar software (Sixth Edition, LASERGENE for Windows & Macintosh, DNASTAR, Inc. Wisconsin, WI, USA) was used to analyze LQ-YT and QL-YT 14-mer peptides. The C-F method and G-R method were used to predict their secondary structures; the K-D method was used to predict their hydrophilicity. The Titration Curve simulation function of the Protean tool was used to analyze LQ-YT and QL-YT peptides. The isoelectric point curves were exhibited.

The far ultraviolet CD spectra of the synthesized 14-mer peptides were recorded using a J-1500 automatic recording spectropolarimeter (JASCO, Tokyo, Japan) at 25 °C. Samples were prepared using phosphate-buffered saline (PBS) at a concentration of l g/L and pH 7.04. Scans were obtained within a range of 185 nm to 260 nm by taking points every 0.1 nm, with an integration time of l s and a 2 nm bandwidth.

### 2.3. Characterization of Fe_3_O_4_-MNPs

The structure of the Fe_3_O_4_-MNPs was analyzed using an X-ray diffractometer (DX-1000, Dandong Fuangyuan Instruments Co. Ltd., Dandong, China) using Cu Ka radiation (k = 1.5418 Å, 40 kV, 80 mA) with a step size of 0.06°. Fe_3_O_4_-MNPs morphology was observed using transmission electron microscopy (TEM, Hitachi H600IV, Tokyo, Japan).

### 2.4. Modification and Characterization of Fe_3_O_4_-MNPs

The modification of Fe_3_O_4_-MNPs was carried out via facile incubation of Fe_3_O_4_-MNPs with peptide solution. Briefly, 20 mg of Fe_3_O_4_-MNPs were pretreated in 10 mL PBS containing 5 mg LQ-YT or QL-YT for 1 h at room temperature, respectively. The suspension was centrifuged to separate particles. After five washes with PBS, the suspension was centrifuged and then dried with vacuum freeze-drying equipment at 37 °C. Precipitates were LQ-YT-modified Fe_3_O_4_-MNPs or QL-YT-modified Fe_3_O_4_-MNPs.

The morphology of the LQ-YT-modified Fe_3_O_4_-MNPs and QL-YT-modified Fe_3_O_4_-MNPs was observed with a field emission scanning electron microscope (SEM, SM-7500F, GEOL, Tokyo, Japan). The elements were evaluated using an energy-dispersive spectrum (EDS, Oxford IETEM100, Carl Zeiss SMT Pte Ltd, Berlin, Germany). The FT-IR spectra were recorded with an FT-IR spectrometer (Nicolet 6700, Thermo Nicole, Waltham, MA, USA), KBr pellet pressing method, and wave number range 500–2000 cm^−1^. The thermal properties were analyzed by thermogravimetry (TG) and differential scanning calorimetry (DSC) (TGA/SDTA851e, METTLER TOLEDO, Zurich, Switzerland) at a heating rate of 10 °C/min in an N_2_ atmosphere.

The stability and zeta potential of the modified Fe_3_O_4_-MNPs in serum were observed. Briefly, 50 μL of different formulations of LQ-YT-modified Fe_3_O_4_-MNPs were added to 1 mL PBS, containing 10% fetal bovine serum (FBS), and then incubated in an incubation shaker (HZ-9210K, CZ Shenglan, Changzhou, China) at 37 °C with gentle oscillation at 30 rpm for 0, 6, and 12 h. The optical density (OD) values were measured at 750 nm using a microplate reader (Microplate Reader 3550, Bio-Rad, Hercules, CA, USA).

The magnetic properties of LQ-YT-modified Fe_3_O_4_-MNPs were investigated using a vibrating sample magnetometer (VSM, Lake shore-7400, Lake Shore Cryotronics, Inc., Columbus, OH, USA).

### 2.5. Identification of Cellular Affinity

The cellular affinities of the modified Fe_3_O_4_-MNPs were identified via immunofluorescence assays. A2780 cells, L929 cells, and HUVECs (1 mL) were inoculated in the 24-well culture plate at the density of 1 × 10^4^. They were then cultured on cover glasses in 24-well plates and fixed with 4% PFA, before being blocked with 2% PBS-BSA triton. Then the cells were incubated with 100 µg/mL of modified Fe_3_O_4_-MNPs for 1 h at room temperature. After that, cells were washed twice with PBS. The cell nuclei were stained with 6-diamidino-2-phenylindole (DAPI) for 8 min at room temperature and then observed with a confocal laser scanning microscope (Leica TCS SP5 MP, Leica Microsystems Vertrieb Gmbh, Wetzlar, Germany).

The cellular affinities of modified Fe_3_O_4_-MNPs were further tested using Prussian blue staining. A2780 cells, L929 cells, and HUVECs (1 mL) were inoculated in the 24-well culture plate at the density of 1 × 10^4^. Suspensions of modified Fe_3_O_4_-MNPs (100 μg/mL) were added to each well. After co-culturing for 4 h, 400 μL paraformaldehyde solution (4%) was added to each well to fix the cells for 30 min. The fixed cells were incubated with 2% potassium ferrocyanide in 2% hydrochloric acid for 30 min. The cells then underwent evaluation using an Olympus IX71 microscope (Olympus, Tokyo, Japan) after the staining solution was removed.

### 2.6. MTT Assay and Scratch Wound Migration Assay

The influence of the modified Fe_3_O_4_-MNPs on the viability of the three kinds of cell lines was determined using the MTT method. A2780 cells, L929 cells, and HUVECs (200 μL) were inoculated in the 96-well culture plate at the density of 1 × 10^4^ overnight and cultured with a concentration gradient of modified Fe_3_O_4_-MNPs (0, 6.25, 12.5, 25, 50, and 100 µg/mL). After 24, 48, and 72 h, 20 μL of 5 mg/mL MTT was added to each well and incubated for 4 h. MTT solution was removed and the resultant formazan crystals formed by viable cells were dissolved in DMSO (150 μL/well). The OD values were measured using a microplate reader (Microplate Reader 3550, Molecular Devices, Sunnyvale, CA, USA) at 490 nm. Cell viability and proliferation were determined in relation to controls.

Scratch wound migration assays were applied to study the influence of modified Fe_3_O_4_-MNPs on cell migration. A2780 cells and HUVECs (2 mL) were inoculated in the 6-well culture plate at the density of 5 × 10^5^ overnight. A straight line was scratched through every well and then cells were inoculated with 50 µg/mL of modified Fe_3_O_4_-MNPs solution without serum for 24 h. The untreated cells were considered as a control. Distances between two sides of the cells were measured using Image J software (V1.8.0, National Institutes of Health, Bethesda, MD, USA).

### 2.7. Statistical Analysis

All data were processed by PASW Statistics 18 (SPSS Inc, Chicago, IL, USA). The results were presented as mean ± SD. Significance was assessed using one-way ANOVA with Bonferroni correction, and the significant differences between the groups were assessed using Student’s t-test and LSD. Statistical significance was defined as P < 0.05.

## 3. Results

### 3.1. Composition and Secondary Structure of Synthesized 14-mer Peptides

The secondary structure of the polypeptide was an important part of the functional domain of the protein, which contained the special information of protein interaction and function. The hydrophobicity difference of the polypeptide had an effect on the folding and high-level conformation, the function, and thermodynamic stability of proteins. At the same time, the surface probability, the flexibility, and antigenicity contained a wealth of biological significance.

The Protean tool of DNAStar software was used to analyze LQ-YT and QL-YT peptides. The C-F method and G-R method were used to predict their secondary structures; the K-D method was used to predict their hydrophilicity. Secondary structures and hydrophilicity prediction results (Figure 1) indicated that the secondary structures of LQ-YT and QL-YT peptides were mainly composed of β-sheets and β-turns, while random coils contributed a little. The relative simple secondary structure and hydrophilicity would make the LQ-YT and QL-YT peptides more likely to participate in the integration platform for the targeting functions.

The Titration Curve simulation function of the Protean tool was used to analyze LQ-YT and QL-YT peptides. The isoelectric points of LQ-YT and QL-YT were pH 8.9, and polypeptides were carried with 0.8 e+ in physiological conditions of pH = 7.2–7.4 (Figure 2a). Positively charged LQ-YT and QL-YT were easy to combine with negatively charged phospholipids and surface molecules of tumor cells.

Figure 2b shows the CD spectra of the LQ-YT and QL-YT 14-mer peptides in PBS buffer at pH 7.04. The strong single negative peaks (appearing at about 210 nm and 216 nm) fit the characteristic protein structure of a β-sheet. The CDNN software peptide secondary structure analysis indicated that LQ-YT was comprised of 41.4% β-sheets, 30.2% β-turns, and 15% random coils; QL-YT was made up of 82.5% β-sheets, 11.1% β-turns, and 4.5% random coils. The CD spectrum results for the LQ-YT and QL-YT peptides were in strong agreement with the software predictions shown in Figure 1. LQ-YT and QL-YT have rather simple and stable secondary structures, which might facilitate the modification of Fe_3_O_4_-MNPs.

### 3.2. Affinity of Synthesized 14-mer Peptides to Fe_3_O_4_-MNPs

The TEM image of the Fe_3_O_4_ powders before modification (Figure 3a_1_) revealed spheroid particles with a mean diameter of 20 nm, and the XRD patterns of the used powders (Figure 3a_2_) showed six obvious diffraction peaks which matched well with the cubic crystal phase Fe_3_O_4_ standard card (PDF No. 75-0449) at 2-theta values of 30.3° (220), 35.7° (311), 43.5° (400), 53.9° (442), 57.5° (511), and 63.2° (440), indicating that the used powders were the cubic Fe_3_O_4_ nanoparticles.

SEM images and EDS analysis of the LQ-YT-modified Fe_3_O_4_-MNPs and QL-YT-modified Fe_3_O_4_-MNPs are shown in Figure 3(b_1_–b_4_). Both LQ-YT- and QL-YT-modified Fe_3_O_4_-MNPs were well dispersed with a uniform size of around 20 nm, indicating that the modifications had little effect on the size of the Fe_3_O_4_-MNPs, but would be propitious to improving the dispersion of Fe_3_O_4_-MNPs. The existence of the different amounts of C and N might be attributed to the fact that the peptides had been conjugated on the surfaces of the Fe_3_O_4_-MNPs, and the amount of conjugated LQ-YT was a little more than that of QL-YT.

The FT-IR absorption spectra of the modified Fe_3_O_4_-MNPs and the constituent Fe_3_O_4_-MNPs, polypeptides, and the physical mixtures of the peptides with Fe_3_O_4_-MNPs further confirmed the interactions of the polypeptides with the Fe_3_O_4_-MNPs (Figure 4a,b). The absorption peaks at 580^−1^ characterized the Fe–O stretching-vibration pattern. The characteristic peaks of the LQ-YT peptide—including amide (I) stretching at 1634 cm^−1^, amide (II) bending at 1538 cm^−1^, and amide (III) stretching at 1456 cm^−1^—appeared distinctly in the spectra of the LQ-YT-modified Fe_3_O_4_-MNPs and in those of the mixture of LQ-YT peptide with Fe_3_O_4_-MNPs. Meanwhile, peaks at 1203 cm^−1^ and 1139 cm^−1^ (the stretching vibration of C=O bonds) clearly appeared in the spectrum of the mixture and almost disappeared in the spectrum of LQ-YT-modified Fe_3_O_4_-MNPs. We speculated that the stretching vibration of the C=O bonds was inhibited by bonding with the Fe atoms in Fe_3_O_4_-MNPs; in other words, the groups bound with the Fe_3_O_4_-MNPs might have been the C=O groups. Similar phenomena were observed in the spectrum of the QL-YT-modified Fe_3_O_4_-MNPs. The disappearance of peaks at 1204 cm^−1^ (the stretching vibration of C=O bonds), 1060 cm^−1^, and 1020 cm^−1^ (the stretching vibration of C–O bonds) indicated that QL-YT peptides were bound to Fe_3_O_4_-MNPs by C–O and C=O. These findings suggested that both LQ-YT and QL-YT exhibited an affinity for Fe_3_O_4_-MNPs and successfully bonded to Fe_3_O_4_-MNPs. The TG–DSC analyses (Figure 4c,d) further suggested that the conjugated proportions of LQ-YT and QL-YT on Fe_3_O_4_-MNPs were 4% (w/w) and 2% (w/w), respectively. The quantity of LQ-YT bound to the surface of Fe_3_O_4_-MNPs was double that of QL-YT, suggesting that β-turns and random coils might increase the affinity of these 14-mer peptides for Fe_3_O_4_-MNPs.

### 3.3. Properties and Performances of Modified Fe_3_O_4_-MNPs

The stability of the 14-mer peptide-modified Fe_3_O_4_-MNPs in serum would be one of the most important factors influencing their fate in vivo. FBS (10%) was always used to simulate physiological conditions when investigating the stability of nanoparticles in serum, by measuring in vitro turbidity. The time variations of turbidities and zeta potentials for LQ-YT-modified Fe_3_O_4_-MNPs in different concentrations are illustrated in Figure 4e. No significant increase in turbidity (indicated by the decrease of OD) was observed after 6 h or 12 h in 10% FBS for all different concentrations. There was no obvious variation in the zeta potential of the LQ-YT-modified MNPs, suggesting that electrostatic repulsion between the negatively charged particles might favor the stable dispersion of nanoparticles.

The hysteresis loops of Fe_3_O_4_-MNPs were traced at room temperature with a VSM. The saturation magnetization (Ms) and the coercivities (Hc) of the Fe_3_O_4_-MNPs before and after modification with the LQ-YT peptide were 80.14 and 76.7 emu/g, and 104.7 and 93.7 Oe, respectively (Figure 4f). The small decreases of Ms and Hc indicated a slight influence of LQ-YT modification on the ferromagnetism of Fe_3_O_4_-MNPs. The results demonstrated that the LQ-YT peptide had little effect on ferromagnetism of Fe_3_O_4_-MNPs. The small reduction of Ms is consistent with the small percentage of added material, as seen from TG–DSC (Figure 4c,d). The small decreases of Hc might be due to different MNPs’ grain size distributions in the measured samples.

### 3.4. Specificity of Modified Fe_3_O_4_-MNPs Bound to A2780 Cells

To verify the specificity of modified Fe_3_O_4_-MNPs bound to A2780 cells, two kinds of negative controls (HUVECs, L929) were included in immunofluorescence and Prussian blue staining assays. 

With the immunofluorescence assays, an objective assessment of differences in intensity levels between the groups could be obtained. Digital images were captured, with the same settings applied for each picture. FITC-labeled fluorescent signals (green) were obviously observed in A2780 cells, HUVECs, and L929 cells cultured with FITC-QL-YT-modified MNPs (Figure 5a–c). In contrast, appreciable fluorescent signals of FITC-LQ-YT-modified MNPs were detected in A2780 cells (Figure 5g), whereas only blue fluorescent signals from nucleus staining were observed in HUVECs and L929 cells (Figure 5h,i). The findings of the immunofluorescence assays suggested that the LQ-YT-modified MNPs could specifically bond to A2780 cells, whereas the QL-YT-modified MNPs nonspecifically bonded to A2780 cells, HUVECs, and L929 cells.

Direct Prussian blue staining could reveal the existence of Fe_3_O_4_-MNPs. After co-incubation of cells with modified MNPs, QL-YT-modified MPs were found around the A2780 cells, HUVECs, and L929 cells (Figure 5d–f). Conversely, the LQ-YT-modified MNPs clearly emerged around the A2780 cells (Figure 5j), but there were scarcely any LQ-YT-modified MNPs around the HUVECs and L929 cells (Figure 5k,l). The Prussian blue staining assay findings further support the specific affinity of LQ-YT-modified MNPs for A2780 cells.

### 3.5. Cytotoxicity of Polypeptide-Modified Fe_3_O_4_-MNPs

The cytotoxicity of polypeptide-modified Fe_3_O_4_-MNPs were assessed by MTT assays. A2780 cells, HUVECs, and L929 cells were treated with QL-YT- and LQ-YT-modified Fe_3_O_4_-MNPs in vitro for 1, 2, and 3 days (Figure 6). After 3 days of treatment in all tested concentrations, LQ-YT-modified Fe_3_O_4_-MNPs significantly inhibited the growth of A2780 tumor cells and significantly promoted the growth of HUVECs and L929 cells (*P* < 0.05). In contrast, QL-YT-modified MNPs were associated with non-significant inhibition of A2780 cells and promotion of L929 cells (*P* > 0.05); they were also associated with significantly accelerated growth of HUVECs (*P* < 0.05). These findings suggested that neither LQ-YT- nor QL-YT-modified Fe_3_O_4_-MNPs affect significant cytotoxicity for normal HUVECs and L929 cells, and that LQ-YT-modified Fe_3_O_4_-MNPs inhibit the growth of A2780 tumor cells. This might be explained by the stronger interactions between the LQ-YT-modified Fe_3_O_4_-MNPs and A2780 cells.

The influence of 14-mer peptide-modified MNPs on cell migration was testified via scratch wound migration assays. At 24 h after scratching, the wound widths associated with the A2780 cells (control) were clearly narrower than those associated with A2780 cells incubated with either LQ-YT- or QL-YT-modified MNPs. Furthermore, the wound widths associated with HUVECs incubated with the two kinds of modified MNPs were found to be similar to the corresponding control (Figure 6). These findings revealed that both types of peptide-modified Fe_3_O_4_-MNPs could inhibit the migration of A2780 tumor cells via interaction with the A2780 cells. Additionally, the modified Fe_3_O_4_-MNPs had no significant effect on the migration of normal HUVECs.

### 3.6. Discussion on Tumor Targeting of Related 14-mer Peptides

Drawing from our previous research on QL-TY and LQ-TY peptides [26], we found significant behavioral differences between the four related 14-mer peptides that were formed by linking the Q-L peptide (the tumor-targeting 7-mer peptide) and T-Y peptide (the Fe_3_O_4_-MNPs affinity 7-mer peptide) in different carbon–nitrogen terminal connection configurations. Except for the LQ-TY peptide, these related 14-mer peptides exhibited affinity for Fe_3_O_4_-MNPs. Meanwhile, both QL-TY- and QL-YT-modified Fe_3_O_4_-MNPs nonspecifically bonded to A2780 cells, HUVECs, and L929 cells, but LQ-YT-modified Fe_3_O_4_-MNPs specifically bonded to A2780 tumor cells.

Although the amino acid compositions of the four 14-mer peptides were exactly the same, there were differences in peptide sequences due to the different carbon–nitrogen terminal connection arrangements. It was reported that C-terminals of amino acids at the external side facilitated the targeting sites of ligands to bond to tumor tissue [27]. In various combinations of Q-L and T-Y 7-mer peptides, the 14-mer peptide LQ-YT was connected in the CN-NC terminal connection configuration, while QL-YT, QL-TY, and LQ-TY peptides were in the NC-NC, NC-CN, and CN-CN configurations, respectively. Among three 14-mer peptides with Fe_3_O_4_-MNPs affinity, only LQ-YT-modified Fe_3_O_4_-MNPs provided the C-terminal of the targeting ligand (L-Q peptide) at the external side. It seemed to be consistent with the above report.

More importantly, the secondary structure of bi-functional peptides might also significantly affect the availability of targeting sites. Secondary structure prediction and CD spectrum analysis revealed that LQ-YT contained more β-turns and random coils. The β-turns and random coils might increase the surface area of LQ-YT relative to the other 14-mer peptides, with the targeting group more likely to be exposed on the surface. These differences might be conducive to exposing the bonding sites on LQ-YT to target cells (ovarian cancer cells). The results of cell immunofluorescence and Prussian blue staining assays also showed that LQ-YT-modified Fe_3_O_4_-MNPs (CN-NC) could specifically bond to and inhibit the growth of A2780 tumor cells. Nevertheless, QL-YT-modified Fe_3_O_4_-MNPs (NC-NC) exhibited inferior targeting specificity to ovarian cancer cells.

## 4. Discussion

In this work, a novel bi-functional 14-mer peptide with targeting affinity for ovarian carcinoma cells and affinity for Fe_3_O_4_-MNPs was designed and synthesized, and then a facile and effective modification method was developed to grant the Fe_3_O_4_-MNPs tumor targeting capabilities. On the basis of the tumor-targeting 7-mer peptide (Q-L) and the other Fe_3_O_4_-MNPs-affinitive 7-mer peptide (T-Y) screened by phage-displayed peptide libraries, four 14-mer complex polypeptides were obtained via linking the Q-L peptide with the T-Y peptide in different carbon–nitrogen terminal connection arrangements. Except for LQ-TY, the related 14-mer peptides exhibited affinity for Fe_3_O_4_-MNPs. Meanwhile, both QL-TY and QL-YT peptide-modified Fe_3_O_4_-MNPs nonspecifically bonded to A2780 tumor cells, HUVECs, and L929 cells, and only LQ-YT-modified Fe_3_O_4_-MNPs specifically bonded to A2780 cells. Secondary structure prediction and CD spectrum analysis revealed that the LQ-YT peptide contained more β-turns and random coils, which are thought to be conducive to increasing polypeptide surface area and exposure of the target group and bonding sites on LQ-YT to external targets. Our findings indicate that the bi-functional 14-mer peptide—obtained via linking the tumor-targeting 7-mer peptide (Q-L) with the Fe_3_O_4_-MNPs-affinitive 7-mer peptide (T-Y) in predetermined configurations—has targeting affinity for ovarian carcinoma cells and affinity for Fe_3_O_4_-MNPs, and that Fe_3_O_4_-MNPs could be fixed with an affinity for ovarian carcinoma cells via a facile modification with this bi-functional 14-mer peptide.

## Figures and Tables

**Figure 1 materials-12-00755-f001:**
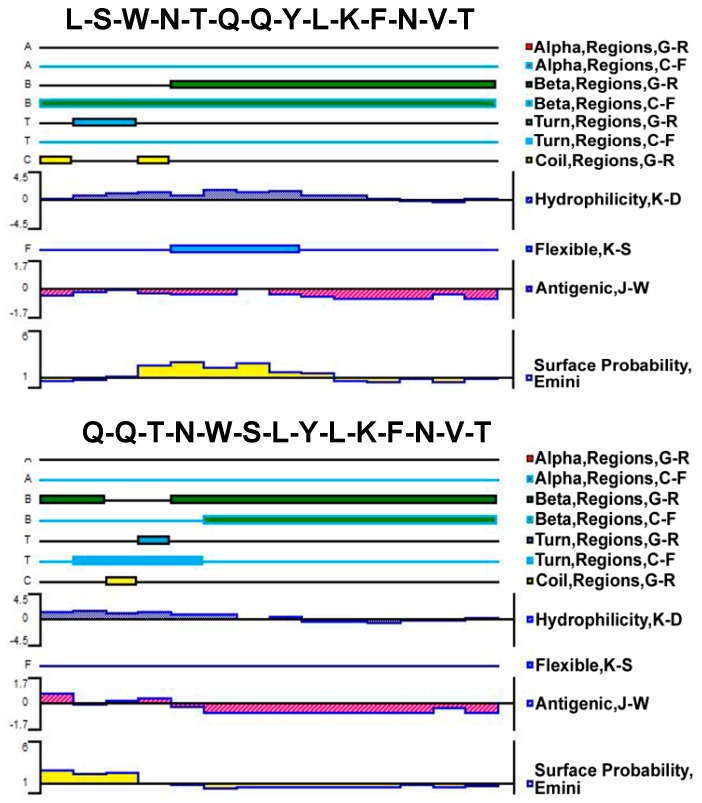
Predicted secondary structures, flexibility plots, hydrophilicity plots, surface probability plots, and antigenicity index for the 14-mer polypeptides.

**Figure 2 materials-12-00755-f002:**
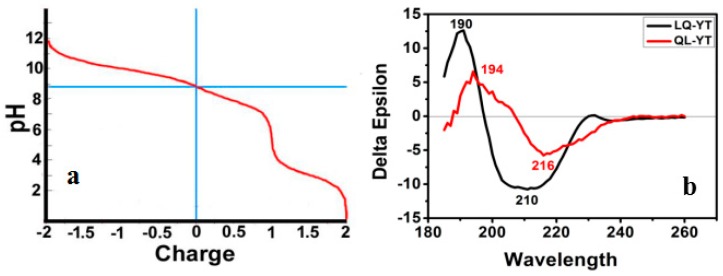
(**a**) Overlapped titration curves for the LQ-YT and QL-YT 14-mer peptides; (**b**) CD spectra of the LQ-YT and QL-YT 14-mer peptides.

**Figure 3 materials-12-00755-f003:**
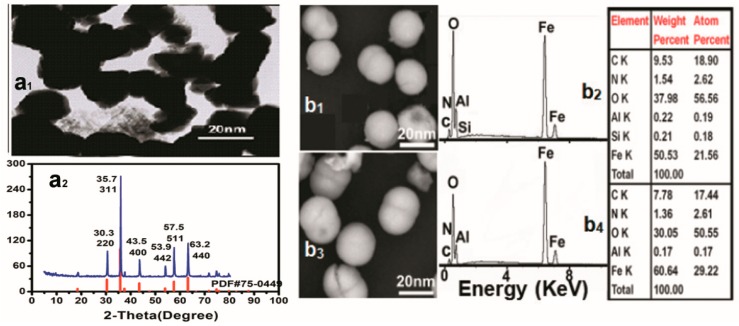
(**a****_1_**) TEM images and (**a****_2_**) XRD patterns of Fe_3_O_4_-MNPs sample as well as (**b_1_**) SEM images and (**b_2_**) EDS spectra of LQ-YT-modified Fe_3_O_4_-MNPs; (**b_3_**) SEM images and (**b_4_**) EDS spectra of QL-YT-modified Fe_3_O_4_-MNPs.

**Figure 4 materials-12-00755-f004:**
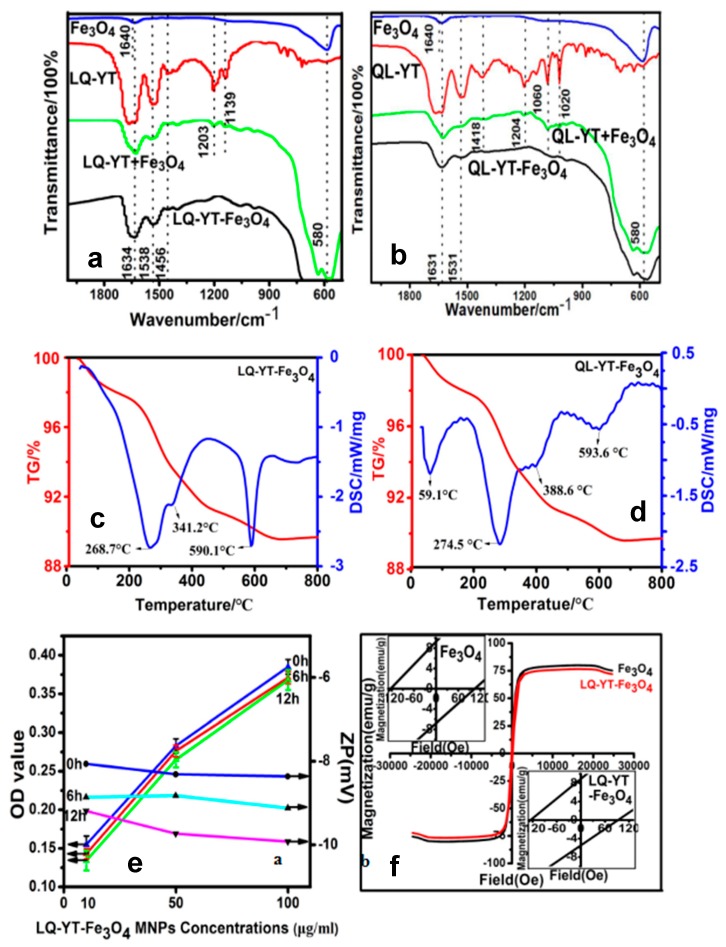
FT-IR spectra curves of (**a**) LQ-YT-modified Fe_3_O_4_-MNPs and (**b**) QL-YT-modified Fe_3_O_4_-MNPs; TG–DSC curves of (**c**) LQ-YT-modified Fe_3_O_4_-MNPs and (**d**) QL-YT-modified Fe_3_O_4_-MNPs; (**e**) serum stabilities curves and (**f**) VSM curves of LQ-YT-modified Fe_3_O_4_-MNPs.

**Figure 5 materials-12-00755-f005:**
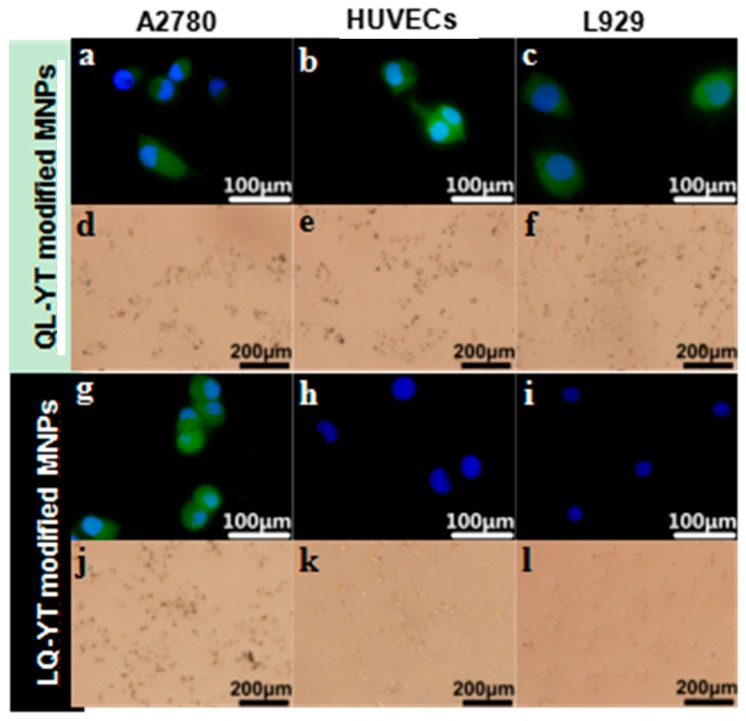
Immunofluorescence images of (**g**–**i**) LQ-YT modified Fe_3_O_4_-MNPs and (**a**–**c**) QL-YT modified Fe_3_O_4_-MNPs; Prussian blue staining images of (**j**–**l**) LQ-YT modified Fe_3_O_4_-MNPs and (**d**–**f)** QL-YT modified Fe_3_O_4_-MNPs.

**Figure 6 materials-12-00755-f006:**
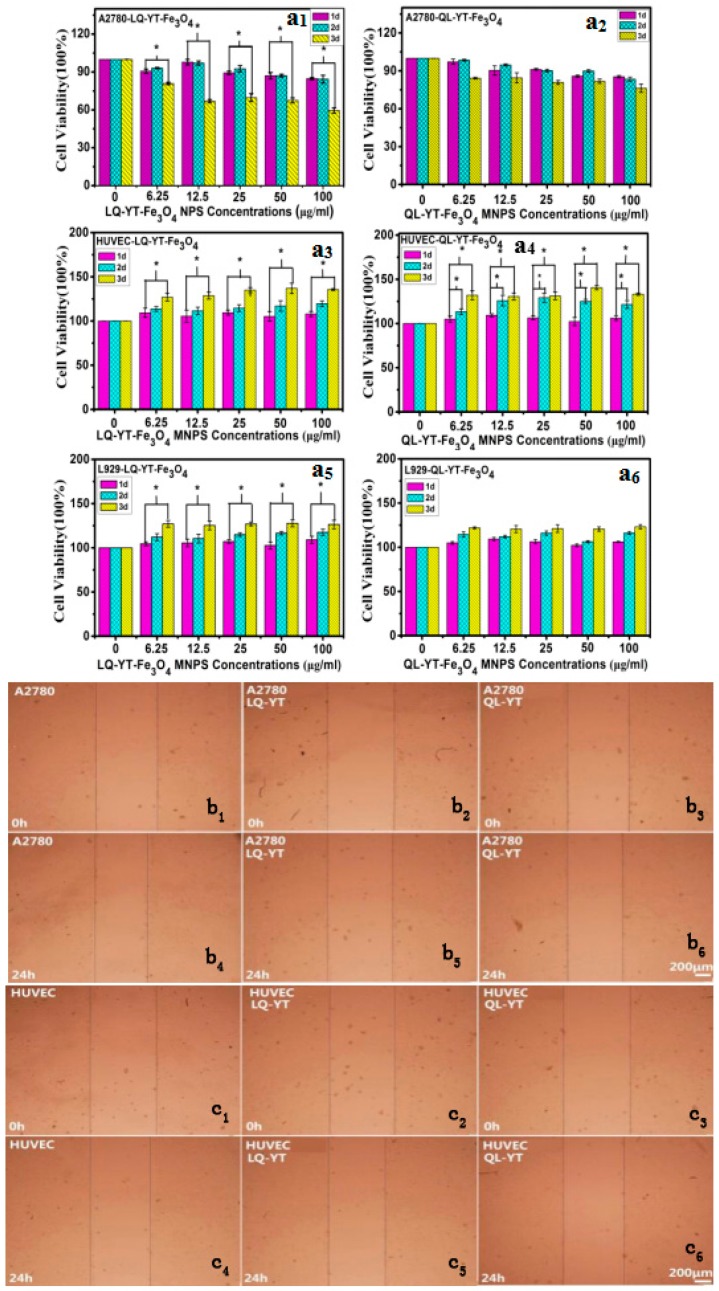
(**a_1_**–**a_6_**) MTT charts of LQ-YT-modified Fe_3_O_4_-MNPs and QL-YT-modified Fe_3_O_4_-MNPs; (**b_1_**–**b_6_**) scratch wound migration images of A2780 incubated with LQ-YT- and QL-YT-modified Fe_3_O_4_-MNPs; (**c_1_**–**c_6_**) scratch wound migration images of HUVECs incubated with LQ-YT- and QL-YT-modified Fe_3_O_4_-MNPs.

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
