# Peer review of "Novel Bi-Functional 14-mer Peptides with Both Ovarian Carcinoma Cells Targeting and Magnetic Fe3O4 Nanoparticles Affinity"

_materials, 2019, doi:10.3390/ma12050755_

Reviewer 1 Report

The manuscript requires extensive editing of English language and style. Some sentences are very difficult to understand. 

The abstract needs some editing. Is not clear in the abstract what are the conclusion of the study. 

Methods are not adequately described. For example for the biological assays authors do not specify the number of cells they used and how they optimised those assays for the different cell types (ovarian cancer cells, fibroblasts and endothelial cells ). 

Although the results are interesting, the authors tested the ability of the modified Fe3O4-MNPs to target ovarian cancer using only one type of ovarian cell lines. It would strength their results the use of at least a second type of ovarian cell  line.

Author Response

Dear Reviewer:

Thank you for your comments concerning our manuscript. Those comments are all valuable and very helpful for revising and improving our paper, as well as the important guiding significance to our researches. We tried our best to improve the manuscript and made some changes in the manuscript.

We hope that the correction will meet with approval.

Point 1: The manuscript requires extensive editing of English language and style. Some sentences are very difficult to understand.

Response 1: Considering the Reviewer’s suggestion, we have re-written this paper according to a native English speaker’s suggestion, the revised portions were marked during revision mode in the paper.

Point 2: The abstract needs some editing. Is not clear in the abstract what are the conclusion of the study.

Response 2: Considering the Reviewer’s suggestion, we have re-written the abstract, the revised portions were marked during revision mode in the paper.

Point 3: Methods are not adequately described. For example, for the biological assays authors do not specify the number of cells they used and how they optimised those assays for the different cell types (ovarian cancer cells, fibroblasts and endothelial cells).

Response 3: As Reviewer suggested that, the methods were not adequately described. Revised portion were marked during revision mode in the paper.

Point 4: Although the results are interesting, the authors tested the ability of the modified Fe3O4-MNPs to target ovarian cancer using only one type of ovarian cell lines. It would strength their results the use of at least a second type of ovarian cell line.

Response 4: It is true as Reviewer suggested that it would strength their results the use of at least a second type of ovarian cell line. We will add this experiment in the following study.

Reviewer 2 Report

The manuscript reports on the study of the synthesis of a novel 14-mer peptide with affinity to magnetic Fe3O4 nanoparticles and to tumors and on the characterization of their structural, morphological, chemical and magnetic properties (before and after modification). Their stability in serum, specificity to target tumor cells and cytocompatibility were also studied. The objective is to circumvent the potential cytotoxicity, high RES clearance in circulation and nonspecific distribution in tissue of Fe3O4 nanoparticles and endow them with tumor targeting, for drug delivery, in particular for ovarian carcinoma cell targeting. The manuscript has original results, but needs minor revisions. I have the following comments:

- Figure 3a2) shows the X-ray diffraction spectrum of the Fe3O4 nanoparticles. On the figure several peaks appear above 2theta ~ 80º seem not to correspond to the ferrite peaks. What is the origin of these peaks ? Are they from secondary phases present in the nanoparticles ?

- Although the remnant magnetization (about 9 emu/g from figure 4f) is small as compared to the saturation magnetization it may still be significant in light of magnetically induced particle aggregation (TEM of figure 3a1 indicates aggregation). Have the authors observed aggregation in their functionalized nanoparticles ?

- The authors refer that the reduction in the saturation magnetization (Ms) and coercive field (Hc) after the addition of LQ-YT is due to the influence of the peptides on the ferromagnetism of the nanoparticles (page 7). However, the added material is not magnetic. As such, it increases the mass of the nanoparticles (more non-magnetic material) but does not contribute to their magnetic moment, effectively reducing the overall magnetization, as was observed. The small reduction of Ms is consistent with the small percentage of added material as seen from TG-DSC. Contrary to what the authors state, this indicates that the magnetic properties of the ferrite NPs themselves were not changed much and are approximately the same after the functionalization (the small change of Hc might be due to different MNPs grain size distributions in the measured samples).

- From figure 4f it is observed that for applied magnetic fields above 20000 Oe the magnetization, which was saturated, starts to decrease, which is odd. What is the origin of this decrease ?

- The y-axis of figure 3a2) should have tick marks to see if the scale is linear or logarithmic. The sentence “It seemed to consist with the above report” should be “It seemed to be consistent with the above report” (page 10).

- The written English is understandable, but should be improved. It is suggested that the authors give it to a native English speaker for improvements.

Author Response

Response to Reviewer 2 Comments

Dear Reviewer:

Thank you for your comments concerning our manuscript. Those comments are all valuable and very helpful for revising and improving our paper, as well as the important guiding significance to our researches. We tried our best to improve the manuscript and made some changes in the manuscript.

We hope that the correction will meet with approval.

Point 1: Figure 3a2 shows the X-ray diffraction spectrum of the Fe3O4 nanoparticles. On the figure several peaks appear above 2theta ~ 80º seem not to correspond to the ferrite peaks. On the figure what is the origin of these peaks? Are they from secondary phases present in the nanoparticles?

Response 1: We are very sorry for our negligence of the figure quality, the diffraction peaks around 2theta ~ 80º of Fe3O4 standard card (PDF No.75-0449) were too short and thin for you to identify. We changed the figure 3a2 to a more clearly one. Several peaks of Fe3O4 nanoparticles samples appear above 2theta ~ 80º correspond to the Fe3O4 standard card peaks.

Point 2: Although the remnant magnetization (about 9 emu/g from figure 4f) is small as compared to the saturation magnetization it may still be significant in light of magnetically induced particle aggregation (TEM of figure 3a1 indicates aggregation). Have the authors observed aggregation in their functionalized nanoparticles?

Response 2: As Reviewer considered that, nanoparticles had small remnant magnetization, which may lead to nanoparticles aggregation. Therefore, we modified the nanoparticles with 14-mer peptides. According to the results of serum stability test (figure 4e), we can observe that there was no obvious aggregation of the modified nanoparticles, indicating that the application performance of nanoparticles had been ameliorated.

Point 3: The authors refer that the reduction in the saturation magnetization (Ms) and coercive field (Hc) after the addition of LQ-YT is due to the influence of the peptides on the ferromagnetism of the nanoparticles (page 7). However, the added material is not magnetic. As such, it increases the mass of the nanoparticles (more non-magnetic material) but does not contribute to their magnetic moment, effectively reducing the overall magnetization, as was observed. The small reduction of Ms is consistent with the small percentage of added material as seen from TG-DSC. Contrary to what the authors state, this indicates that the magnetic properties of the ferrite NPs themselves were not changed much and are approximately the same after the functionalization (the small change of Hc might be due to different MNPs grain size distributions in the measured samples).

Response 3: It is true as Reviewer suggested that the small change of Hc might be due to different MNPs grain size distributions in the measured samples. We had studied comment carefully and made correction. Revised portion are marked in red in the paper.

Point 4: From figure 4f it is observed that for applied magnetic fields above 20000 Oe the magnetization, which was saturated, starts to decrease, which is odd. What is the origin of this decrease?

Response 4: The results of VSM (figure 4f) came from the results of objective experiment. We tried to search the literatures for possible reasons of magnetization decrease, but failed. Since this was the result of multiple tests, it may be due to the unstable experiment equipment. We will try to find out the reasons in the following study.

Point 5: The y-axis of figure 3a2 should have tick marks to see if the scale is linear or logarithmic. The sentence “It seemed to consist with the above report” should be “It seemed to be consistent with the above report” (page 10).

Response 5: We are very sorry for our incorrect writing. We had corrected the wrong sentence. Revised portion were marked in red in the paper. The y-axis of figure 3a2 had been added in the paper.

Point 6: The written English is understandable, but should be improved. It is suggested that the authors give it to a native English speaker for improvements.

Response 6: Considering the Reviewer’s suggestion, we had re-written this paper according to a native English speaker’s suggestion, the revised portions were marked in revision mode in the paper.

Reviewer 3 Report

The present manuscript entitled " Novel Bi-functional 14-mer Peptides with both Ovarian Carcinoma Cells Targeting and Magnetic Fe3O4 Nanoparticles Affinity”, describes the design and synthesis of a novel bi-functional 14-mer peptide with targeting to ovarian carcinoma cells and magnetic Fe3O4 nanoparticles affinity. Moreover Fe3O4-MNPs with tumor targeting via the conjugation with the bi-functional peptides were obtained. It is a topic of interest to the researchers in the related areas, but the results should be discussed more and the paper needs significant improvement before the publication. My detailed comments are as follows:

Line 36-37 More recent references should be included.

Line 110 Materials and cell culture section should be moved above as paragraph 2.1.

Line 100 Peptides synthesis is not clear. Synthesis procedure should be included in the text.

Line 127 Fe3O4MNPs were observed by TEM while LQ-YT modified Fe3O4-MNPs and QL-YT modified Fe3O4-MNPs were observed by SEM.  Why were two different techniques used? For a suitable comparison TEM image of Fe3O4-MNPs should be compared with TEM image of modified Fe3O4-MNPs. TEM and SEM images of both samples should be provided. I further suggest you to include dimensional analysis (e.g. DLS analysis). Which the polydispersity of the sample?

Line 133 Freeze-drying conditions should be specified.

Line 138 FT-IR conditions should be specified.

Line 140 At each time point tested, OD values (at 750 nm) of only medium without NPs should be evaluated as control.

Line 150 Number of cells and seeding volume should be included in the text.

Line 151 Why did you fix the cells before treating with Fe3O4MNPs? Normally, cells are treated with NPs firstly and then they are fixed by PTA. After PTA fixing, cells are died, and they are not able to internalize the NPs.

Line 156 Number of cells and seeding volume should be included in the text.

Line 166 Number of cells and seeding volume should be included in the text.

Line 170  "Control" means untreated cells? It should be specified in the text.

Figure 1 – Quality of image should be improved. Figure legend should be revised and make more clear.

Lines 189-201 Results and figure 1 should be discussed more.  It should be rewritten in order to make it more comprehensible. LQ-YT and QL-YT were synthetized, was the secondary structure determined for all samples? if no, why?

Figure 3a – Quality of image should be improved, TEM data is not informative: the roughly spherical shape is not appreciable. Scale bar is omitted.

Lines 238- 243 It was speculated that C=O might be the groups bound the Fe atoms in Fe3O4-MNPs. Are there any evidences in literature? Can you prove this hypothesis? Interaction of peptide and Fe3O4MNPs needs to be explained at least schematically.

Line 246 Please, indicate if % refers to w/w or mol/mol.

Line 246 Quantitative HPLC or UV methods should be included to quantify peptides associated to Fe3O4-MNPs.

Figure 4 Quality of image should be improved, Figure 4e legend should be revised to make clearer the list of samples. Figure 5e should be made clearer.

Figure 5 Quality of image should be improved.

Line 246 Are there evidences about the cytotoxicity of Fe3O4MNPs? Why should LQYT modified Fe3O4-MNPs inhibit the A2780 cells viability while they should promote the HUVECs and L929 viability? Should the interaction between peptide and cells cause the cell dead?

Figure 6 (b1-c6) Quality of image should be improved, the magnification should be increased because cell morphology is not appreciable.

Author Response

Response to Reviewer 3 Comments

Dear Reviewer:

Thank you for your comments concerning our manuscript. Those comments are all valuable and very helpful for revising and improving our paper, as well as the important guiding significance to our researches. We tried our best to improve the manuscript and made some changes in the manuscript.

We have re-written this paper according to a native English speaker’s suggestion, the revised portions were marked in revision mode in the paper.

We hope that the correction will meet with approval.

Point 1: Line 36-37 More recent references should be included.

Response 1: As Reviewer suggested that, we had added two recent references in red in the paper.

Point 2: Line 110 Materials and cell culture section should be moved above as paragraph 2.1.

Response 2: We are very sorry for our negligence of the logical order, we had moved the Materials and cell culture section above as paragraph 2.1. Revision is red in the paper.

Point 3: Line 100 Peptides synthesis is not clear. Synthesis procedure should be included in the text.

Response 3: TVNFKLY and QQTNWSL 7-mer peptides were synthesized into two 14-mer peptides LQ-YT and QL-YT. LQ-YT and QL-YT 14-mer peptides were synthesized by the Shanghai Bootech Bioscience Technology Co. Ltd (Shanghai, China) according to our authorization.

Point 4: Line 127 Fe3O4-MNPs were observed by TEM while LQ-YT modified Fe3O4-MNPs and QL-YT modified Fe3O4-MNPs were observed by SEM. Why were two different techniques used? For a suitable comparison TEM image of Fe3O4-MNPs should be compared with TEM image of modified Fe3O4-MNPs. TEM and SEM images of both samples should be provided. I further suggest you to include dimensional analysis (e.g. DLS analysis). Which the polydispersity of the sample?

Response 4:

      Why were two different techniques used?

It is true as reviewer suggested that TEM and SEM images of both samples should be better. We will supply this experiment in the following study according to your suggestion.

  Which the polydispersity of the sample?

As Reviewer suggested that, DSL analysis was necessary. This test not only measure the mean particle size, but also can get PDI (polydispersity index, particle size distribution) results. We will supply this experiment in the following study.

Point 5: Line 133 Freeze-drying conditions should be specified.

Response 5: We are very sorry for our negligence, revision is red in the paper.

Point 6: Line 138 FT-IR conditions should be specified.

Response 6: We have added the FT-IR conditions; revision is red in the paper.

Point 7: Line 140 At each time point tested, OD values (at 750 nm) of only medium without NPs should be evaluated as control.

Response 7: It is true as reviewer suggested that at each time point test, OD values (at 750 nm) of only medium without NPs should be evaluated as control. We will improve this experiment in the following tests.

Point 8: Line 150 Number of cells and seeding volume should be included in the text.

Response 8: We are very sorry for our negligence, revision is red in the paper.

Point 9: Line 151 Why did you fix the cells before treating with Fe3O4-MNPs? Normally, cells are treated with NPs firstly and then they are fixed by PTA. After PTA fixing, cells are died, and they are not able to internalize the NPs.

Response 9: It is true as reviewer suggested that after PTA fixing, cells were died. We will change this experiment method order in the following study.

Point 10: Line 156 Number of cells and seeding volume should be included in the text.

Response 10: We have added the number of cells, revision is red in the paper.

Point 11: Line 166 Number of cells and seeding volume should be included in the text.

Response 11: We have added the number of cells, revision is red in the paper.

Point 12: Line 170 "Control" means untreated cells? It should be specified in the text.

Response 12: Control" means untreated cells, and it has been specified in the text, revision is red in the paper.

Point 13: Figure 1 – Quality of image should be improved. Figure legend should be revised and make more clearly.

Response 13: We are very sorry for our negligence of the figure quality, we have changed the figure to a more clearly one. Figure legend revision is red in the paper.

Point 14: Lines 189-201 Results and figure 1 should be discussed more. It should be rewritten in order to make it more comprehensible. LQ-YT and QL-YT were synthetized, was the secondary structure determined for all samples? If no, why?

Response 14: Results and figure 1 section have been re-written, revision is red in the paper. Two 7-mer peptides can synthesize four 14-mer peptides. The amino acid sequences of the four 14-mer peptides were different. This will influence the secondary structures and hydrophilic characteristics, as shown in figure 1.

Point 15: Figure 3a – Quality of image should be improved, TEM data is not informative: the roughly spherical shape is not appreciable. Scale bar is omitted.

Response 15: We have changed the Figure 3a to a more clearly one, revision is in the paper.

Point 16: Lines 238- 243 It was speculated that C=O might be the groups bound the Fe atoms in Fe3O4-MNPs. Are there any evidences in literature? Can you prove this hypothesis? Interaction of peptide and Fe3O4-MNPs needs to be explained at least schematically.

Response 16:

You, F.; Yin, G.; Pu, X.; Li, Y.; Hu, Y.; Huang, Z.; Liao, X.; Yao, Y.; Chen, X., Biopanning and characterization of peptides with Fe3O4 nanoparticles-binding capability via phage display random peptide library technique. Colloids Surf B Biointerfaces 2016, 141, 537-545.

 This article said that:

The disappearance of some peaks in FT-IR absorption spectra showed that the vibration of C=O or C-O bonds was limited, which meant that the hydroxyl groups and carboxyl groups interacted with Fe3O4 NPs. This demonstrated that the probable mechanism of selective binding of peptides to Fe3O4 might mainly be the Pearson hard acid-hard base interaction via the specific electrostatic attractions.

This picture was in this article:

Point 17: Line 246  Please, indicate if % refers to w/w or mol/mol.

Response 17: We have indicated that % refers to w/w, revision is red in the paper.

Point 18: Line 246 Quantitative HPLC or UV methods should be included to quantify peptides associated to Fe3O4-MNPs.

Response 18: It is true as reviewer suggested that quantitative HPLC or UV methods should be included. We will conduct the Quantitative HPLC or UV experiments in the following study.

Point 19: Figure 4 Quality of image should be improved, Figure 4e legend should be revised to make clearer the list of samples. Figure 5e should be made clearer.

Response 19: We have changed the figures (Figure 4 and Figure 5) to more clearly ones. Revision is in the paper.

Point 20: Figure 5 Quality of image should be improved.

Response 20: We have changed the figure (Figure 5) to a more clearly one. Revision is in the paper.

Point 21: Line 246 Are there evidences about the cytotoxicity of Fe3O4-MNPs? Why should LQ-YT modified Fe3O4-MNPs inhibit the A2780 cells viability while they should promote the HUVECs and L929 viability? Should the interaction between peptide and cells cause the cell dead?

Response 21: The research results of Master Yucan Li who were in the same research group with us showed that Q-L 7-mer peptide had significant inhibition on the growth of A2780 cells, while slightly promoted the proliferation of HUVECs and L929 cells. In our reserch, 14-mer peptides were designed and synthesized through combining Q-L peptide and T-Y peptide in predetermined manners. The inhibition and facilitation of LQ-YT modified Fe3O4-MNPs may be due to the inhibition and facilitation of Q-L 7-mer peptide. More research should be done to figure out more complicated and essential reasons.

Point 22: Figure 6 (b1-c6) Quality of image should be improved, the magnification should be increased because cell morphology is not appreciable.

Response 22: We had changed the figure to more clearly ones. Revision is in the paper.

Round  2

Reviewer 1 Report

The Authors have addressed all my comments and they have improved significantly the manuscript. I suggest to publish the paper in the present from. 

Reviewer 3 Report

The manuscript has been improved and now warrants publication in Materials